# A Neural-preconditioned Poisson Solver for Mixed Dirichlet and Neumann Boundary Conditions

## Abstract

We introduce a neural-preconditioned iterative solver for Poisson equations with mixed boundary conditions. The Poisson equation is ubiquitous in scientific computing: it governs a wide array of physical phenomena, arises as a subproblem in many numerical algorithms, and serves as a model problem for the broader class of elliptic PDEs. The most popular Poisson discretizations yield large sparse linear systems. At high resolution, and for performance-critical applications, iterative solvers can be advantageous for these—but only when paired with powerful preconditioners. The core of our solver is a neural network trained to approximate the inverse of a discrete structured-grid Laplace operator for a domain of arbitrary shape and with mixed boundary conditions. The structure of this problem motivates a novel network architecture that we demonstrate is highly effective as a preconditioner even for boundary conditions outside the training set. We show that on challenging test cases arising from an incompressible fluid simulation, our method outperforms state-of-the-art solvers like algebraic multigrid as well as some recent neural preconditioners.

## 1 Introduction

The solution of linear systems of equations involving discrete Laplace operators is the bottleneck in many engineering and scientific applications. These large, symmetric positive definite and sparse systems of equations are notoriously ill-conditioned. Fast Fourier Transforms (Cooley & Tukey, 1965) are optimal for these problems when discretized over trivial geometric domains, however they are not applicable for practical domain shapes. Direct methods like Cholesky factorization (Golub & Loan, 2012) resolve conditioning issues, but suffer from loss of sparsity/fill-in and are prohibitively costly in practice when per-time-step refactoring is necessary (*e.g.*, with changing domain shape). Iterative methods like preconditioned conjugate gradient (PCG) (Saad, 2003) and multigrid (Brandt, 1977) can achieve good performance, however an optimal preconditioning strategy is not generally available, and though multigrid can guarantee modest iteration counts, computational overhead associated with solver creation and other per-iteration costs can dominate runtimes in practice. Unfortunately, there is no clear algorithmic solution.

Recently, machine learning techniques have shown promise for these problems. Tompson et al. (2017) showed that a network (FluidNet) can be used to generate an approximate inverse across domain shapes, albeit only with Neumann boundary conditions. Kaneda et al. (2023) developed DCDM (Deep Conjugate Direction Method), which improves on this approach by using a similar network structure and an iterative technique where gradient descent in the matrix norm of the error is preconditioned with a neural network. While their approach is similar to PCG, the nonlinearity of their approximate inverse required a generalization of the PCG method which proved effective. We build on this approach and generalize it to domains with mixed Dirichlet and Neumann boundary conditions. Notably, these problems arise in simulating free-surface liquid flows. The DCDM approach cannot handle these cases, however we show that a novel, more lightweight network structure can be used in DCDM's iterative formalism that is both linear and capable of handling mixed boundary conditions over time-varying fluid domains. Furthermore, we show that this structure drastically improves performance over that in DCDM. We design our

network structure to represent the dense nature of the inverse of a discrete Laplacian matrix. That is, the inverse matrix for a discrete Laplace operator has the property that local perturbations anywhere in the domain have non-negligible effects at all other points in the domain. Our network structure uses a hierarchy of grid scales to improve the resolution of this behavior over what is possible with the DCDM structure. In effect, the process of transferring information across the hierarchy from fine grid to increasingly coarse grids and back again facilitates rapid propagation of information across the domain. This structure has similarities with multigrid, however there are some important differences. We incorporate the effects of the Dirichlet and Neumann conditions at irregular boundaries with a novel convolution design. Specifically, we use stencils that learn spatially varying weights based on a voxel's proximity to the boundary and the boundary condition types encoded there.

Although our approximate inverses are linear (unlike the DCDM preconditioner) we still adopt the DCDM iterative formalism. We do this because we cannot guarantee that our neural network produces a symmetric and positive definite approximate inverse as required for standard PCG. It is possible to use a flexible PCG technique (Golub & Ye, 1999) in this case though (as in (Bouwmeester et al., 2015)), however we show that the matrix-orthogonal gradient descent iteration in DCDM provides superior results. We show that our network outperforms state-of-the-art preconditioning strategies, including DCDM, FluidNet, algebraic multigrid and incomplete Cholesky. We perform our comparison across a number of representative free-surface liquid and fluid flow problems. To promote reproducibility we have released our full code and a link to our pretrained model at `https://anonymous.4open.science/r/MLPCG-2102`.

## 2 RELATED WORK

Many recent approaches leverage machine learning techniques to accelerate numerical linear algebra computations. Ackmann et al. (2020) use supervised learning to compute preconditioners from fully-connected feed-forward networks in semi-implicit time stepping for weather and climate models. Sappl et al. (2019) use convolutional neural networks (CNNs) to learn banded approximate inverses for discrete Poisson equations arising in incompressible flows discretized over voxelized spatial domains. However, their loss function is the condition number of the preconditioned operator which is prohibitively costly at high resolution. Özbay et al. (2021) also use CNN to approximate solutions to Poisson problems arising in incompressible flow discretized over voxelized domains, however they do not learn a preconditioner and their approach only supports two-dimensional square domains. Our approach is most similar to those of Tompson et al. (2017) and Kaneda et al. (2023) who also consider discrete Poisson equations over voxelized fluid domains, however our lighter-weight network outperforms them and generalizes to a wider class of boundary conditions. Li et al. (2023) build on the approach of Sappl et al. (2019), but use a more practical loss function based on the supervised difference between the inverse of their preconditioner times a vector and its image under the matrix under consideration. Their preconditioner is the product of easily invertible, sparse lower triangular matrices. Notably, their approach works on discretizations over unstructured meshes. Götz & Anzt (2018) learn Block-Jacobi preconditioners using deep CNNs. The choice of optimal blocking is unclear for unstructured discretizations, and they use machine learning techniques to improve upon the selection.

Various works use hybrid deep learning/multigrid techniques. For example, the UNet (Ronneberger et al., 2015) and MSNet architectures (Mathieu et al., 2016) are similar to a multigrid V-cycle in terms of data flow, as noted by Cheng et al. (2021) and Azulay & Treister (2023). Cheng et al. (2021) use the multi-scale network architecture MSNet to approximate the solution of Poisson equations arising in plasma flow problems. However, they only consider flows over a square domain in 2D. Azulay & Treister (2023) note the similarity between the multi-scale UNet architecture and a multigrid V-cycle. They use this structure to learn preconditioners for the solution of heterogeneous Helmholtz equations. Eliasof et al. (2023) also use a multigrid-like architecture for a general class of problems. Huang et al. (2023) use deep learning to generate multigrid smoothers at each grid resolution that effectively smooth high frequencies: CNNs generate the smoothing stencils from matrix entries at each level in the multigrid hierarchy. This is similar to our boundary-condition-dependent stencils, however we note that our network is lighter-weight and

allowed to vary at a larger scale during learning. Furthermore, optimal stencils are known for the problems considered in this work, and we provide evidence that our solvers outperforms them.

## 3 MOTIVATION: INCOMPRESSIBLE FLUIDS WITH MIXED B.C.S

While our solver architecture can be applied to any Poisson equation discretized on a structured grid, our original motivation was to accelerate a popular method for incompressible inviscid fluid simulation based on the splitting scheme introduced by Chorin (1967). The fluid's velocity $\mathbf{u}(\mathbf{x}, t)$ is governed by the incompressible Euler equations:

$$\rho \left( \frac{\partial \mathbf{u}}{\partial t} + (\mathbf{u} \cdot \nabla)\mathbf{u} \right) + \nabla p = \mathbf{f}^{\text{ext}} \quad \text{s.t.} \quad \nabla \cdot \mathbf{u} = 0 \quad \text{in } \Omega, \tag{1}$$

where $\Omega$ is the domain occupied by fluid, pressure $p$ is the Lagrange multiplier for the incompressibility constraint $\nabla \cdot \mathbf{u} = 0$, $\rho$ is the mass density, and $\mathbf{f}^{\text{ext}}$ accounts for external forces like gravity. These equations are augmented with initial conditions $\mathbf{u}(\mathbf{x}, 0) = \mathbf{u}^0(\mathbf{x})$ and $\rho(\mathbf{x}, 0) = \rho^0$ as well as the boundary conditions discussed in Section 3.1. Incompressibility implies that the initial homogeneous mass density is conserved throughout the simulation ($\rho \equiv \rho^0$).

Chorin's scheme employs finite differences in time and splits the integration from time $t^n$ to $t^{n+1} = t^n + \Delta t$ into two steps. First, a provisional velocity field $\mathbf{u}^*$ is obtained by an *advection step* that neglects the pressure and incompressibility constraint:

$$\frac{\mathbf{u}^* - \mathbf{u}^n}{\Delta t} + (\mathbf{u}^n \cdot \nabla)\mathbf{u}^n = \frac{1}{\rho^0} \mathbf{f}^{\text{ext}}. \tag{2}$$

Second, a *projection step* obtains $\mathbf{u}^{n+1}$ by eliminating divergence from $\mathbf{u}^*$:

$$-\nabla \cdot \frac{1}{\rho^0} \nabla p^{n+1} = -\frac{1}{\Delta t} \nabla \cdot \mathbf{u}^*, \tag{3}$$

$$\frac{\mathbf{u}^{n+1} - \mathbf{u}^*}{\Delta t} = -\frac{1}{\rho^0} \nabla p^{n+1}. \tag{4}$$

Equations 2-4 hold inside $\Omega$, and we have deferred discussion of boundary conditions to Section 3.1. The bottleneck of this full process is (3), which is a Poisson equation since $\rho^0$ is spatially constant.

### 3.1 BOUNDARY CONDITIONS

Our primary contribution is handling both Neumann and Dirichlet boundary conditions for the Poisson equation. We assume the computational domain $\mathcal{D}$ is decomposed into $\mathcal{D} = \Omega \cup \Omega_a \cup \Omega_s$, as sketched in the inset, where $\Omega_a$ denotes free space and $\Omega_s$ the region filled with solid. This decomposition induces a partition of the fluid boundary $\partial\Omega = \Gamma_n \cup \Gamma_d$. Boundary $\Gamma_n$ represents the fluid-solid interface as well as the intersection $\partial\Omega \cap \partial\mathcal{D}$ (*i.e.*, the region outside $\mathcal{D}$ is treated as solid); on it a free-slip boundary condition is imposed: (1), $\mathbf{u}(\mathbf{x}, t) \cdot \hat{\mathbf{n}}(\mathbf{x}) = u_n^\Gamma(\mathbf{x}, t)$, where $\hat{\mathbf{n}}$ denotes the outward-pointing unit surface normal. This condition on $\mathbf{u}$ translates via (4) into a Neumann condition on (3):

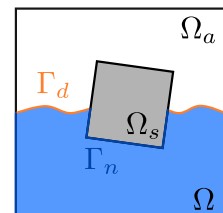

$$\hat{\mathbf{n}} \cdot \nabla p^{n+1} = \frac{\rho_0}{\Delta t} (\hat{\mathbf{n}} \cdot \mathbf{u}^* - u_n^\Gamma) \quad \text{on } \Gamma_n. \tag{5}$$

Free-surface boundary $\Gamma_d$ represents the interface between the fluid and free space. Ambient pressure $p_a$ then imposes on (3) a Dirichlet condition $p^{n+1} = p_a$ on $\Gamma_d$. In our examples, we set $p_a = 0$.

The Dirichlet conditions turn out to make solving (3) fundamentally more difficult: while the DCDM paper Kaneda et al. (2023) discovered that a preconditioner blind to the domain geometry and trained solely on an empty box is highly effective for simulations featuring pure Neumann conditions, the same is not true for Dirichlet (see Figure 5).

### 3.2 SPATIAL DISCRETIZATION

We discretize the full domain $\mathcal{D}$ using a regular marker-and-cell (MAC) staggered grid with $n_c$ cubic elements Harlow (1964). The disjoint subdomains $\Omega$, $\Omega_a$, and $\Omega_s$ are each represented by a per-cell

rasterized indicator field; these are collected into a 3-channel image, stored as a tensor $\mathcal{I}$. In the case of a 2D square with $n_c = N^2$, this tensor is of shape $(3, N, N)$, and summing along the first index yields a single-channel image filled with ones.

Velocities and forces are represented at the *corners* of this grid, and for smoke simulations the advection step (2) is implemented using an explicit semi-Lagrangian method (Stam, 1999; Robert, 1981). For free-surface simulations, advection is performed by interpolating fluid velocities from the grid onto particles responsible for tracking the fluid state, advecting those particles, and then transferring their velocities back to the grid. In our examples, we use a PIC/FLIP blend transfer scheme with a 0.99 ratio (Zhu & Bridson, 2005).

Pressure values are stored at element *centers*, and the Laplace operator in (3) is discretized into a sparse symmetric matrix $A^{\mathcal{I}} \in \mathbb{R}^{n_c \times n_c}$ using the standard second-order accurate finite difference stencil (with 5 points in 2D and 7 in 3D) but with modifications to account for Dirichlet and Neumann boundary conditions: stencil points falling outside $\Omega$ are dropped, and the central value (*i.e.*, the diagonal matrix entry) is determined as the number of neighboring cells belonging to either $\Omega$ or $\Omega_a$. Examples of these stencils are visualized in 2D in the inset. Rows and columns corresponding to cells outside $\Omega$ are left empty, meaning $A^{\mathcal{I}}$ typically has a high-dimensional nullspace. These empty rows and columns are removed before solving, obtaining a smaller positive definite matrix $\tilde{A}^{\mathcal{I}} \in \mathbb{R}^{n_f \times n_f}$, where $n_f$ is the number of fluid cells.

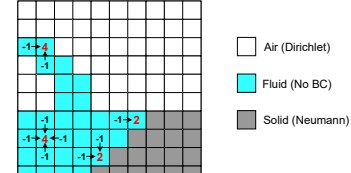

The right-hand side of (3) is discretized using the standard MAC divergence finite difference stencil into a vector $\mathbf{b} \in \mathbb{R}^{n_c}$, which also receives contributions from the Neumann boundary. Entries of this vector corresponding to cells outside $\Omega$ are removed to form right-hand side vector $\tilde{\mathbf{b}} \in \mathbb{R}^{n_f}$ of the reduced linear system representing the discrete Poisson equation:

$$\tilde{A}^{\mathcal{I}} \tilde{\mathbf{x}} = \tilde{\mathbf{b}}, \tag{6}$$

where $\tilde{\mathbf{x}} \in \mathbb{R}^{n_f}$ collects the fluid cells' unknown pressure values (a discretization of $p^{n+1}$).

The constantly changing domains and boundary conditions of a typical fluid simulation mean traditional preconditioners for (6) like multigrid or incomplete Cholesky, as well as direct sparse Cholesky factorizations, need to be *rebuilt at every frame*. This prevents their high fixed costs from being amortized across frames and means they struggle to outperform a highly tuned GPU implementation of unpreconditioned CG. This motivates our neural-preconditioned solver which, after training, instantly adapts to arbitrary subdomain shapes encoded in $\mathcal{I}$.

# 4 NEURAL-PRECONDITIONED STEEPEST DESCENT WITH ORTHOGONALIZATION

Our neural-preconditioned solver combines a carefully chosen iterative method (Section 4.1) with a preconditioner based on a novel neural network architecture (Section 4.2.1) inspired by multigrid.

## 4.1 ALGORITHM

For symmetric positive definite matrices $A$ (like the discrete Laplacian $\tilde{A}^{\mathcal{I}}$ from (6)), the preconditioned conjugate gradient (PCG) algorithm (Shewchuk, 1994) is by far the most efficient iterative method for solving linear systems $A\mathbf{x} = \mathbf{b}$ when an effective preconditioner is available. Unfortunately, its convergence rate is known to degrade when the preconditioner itself fails to be symmetric, as is the case for our neural preconditioner. Bouwmeester et al. (2015) have shown that good convergence can be recovered for nonsymmetric multigrid preconditioners using the "flexible PCG" variant at the expense of an additional dot product. However, this variant turns out to perform suboptimally with our neural preconditioner, as shown in Table 1. Instead, we adopt the preconditioned steepest descent with orthogonalization (PSDO) method proposed in Kaneda et al. (2023), which was shown to perform well even for their nonlinear preconditioning operator.

The PSDO algorithm can be understood as a modification of standard CG that replaces the residual with the preconditioned residual as the starting point for generating search directions and, consequently, cannot enjoy many of the simplifications baked into the traditional algorithm. Most seri-

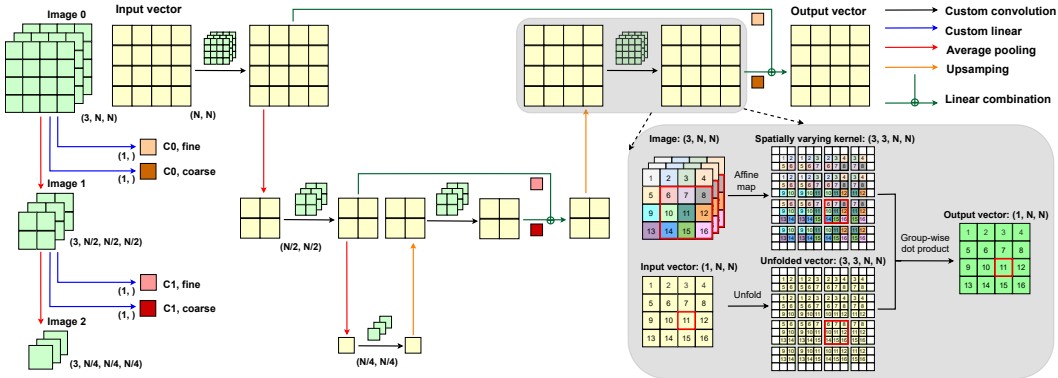

Figure 1: Our network architecture sketched for a 2D preconditioner with $\mathcal{L} = 3$ levels.

ously, $A$-orthogonalizing against only the previous search direction no longer suffices to achieve $A$-orthogonality to all past steps. Therefore, iteration $k$ of PSDO obtains its step direction $\mathbf{d}_k$ by explicitly $A$-orthogonalizing the preconditioned residual against the last $n_{\text{ortho}}$ directions (where $n_{\text{ortho}}$ is a tunable parameter) and determines step length $\alpha_k$ with an exact line search. PSDO reduces to standard preconditioned steepest descent (PSD) when $n_{\text{ortho}} = 0$, and it is mathematically equivalent to unpreconditioned CG when $n_{\text{ortho}} \geq 1$ and the identity operator is used as the preconditioner. In the case of a symmetric preconditioner $P = LL^\top$, PSDO differs from PCG by taking steps that are $A$-orthogonal rather than $LAL^\top$-orthogonal. When combined with our neural preconditioner, we call this algorithm NPSDO, presented formally in Algorithm 1 in the appendix. We empirically determined $n_{\text{ortho}} = 2$ to perform well, and we use this value in all reported experiments.

## 4.2 NEURAL PRECONDITIONER

The ideal preconditioner for all iterative methods described in Section 4.1 is the exact inverse $A^{-1}$; with it, each method would converge to the exact solution in a single step. Of course, the motivation for using an iterative solver is that inverting or factorizing $A$ is too costly (Figure 6), and instead we must seek an inexpensive approximation of $A^{-1}$. Examples are incomplete Cholesky, which does its best to factorize $A$ with a limited computational budget, and multigrid, which applies one or more iterations of a multigrid solver.

Our method approximates the map $\mathbf{r} \mapsto A^{-1}\mathbf{r}$ by our neural network $\mathcal{P}^{\text{net}}(\mathcal{I}, \mathbf{r})$. Departing from recent works like Kaneda et al. (2023), we use a novel architecture that both substantially boosts performance on pure-Neumann problems and generalizes to the broader class of Poisson equations with mixed boundary conditions by considering geometric information from $\mathcal{I}$. The network performs well on 2D or 3D Poisson equations of varying sizes, but to simplify the exposition, our figures and notation describe the method on small square grids of size $N \times N$.

We note that Algorithm 1 runs on linear system $\tilde{A}^{\mathcal{I}}\tilde{\mathbf{x}} = \tilde{\mathbf{b}}$, featuring vectors of smaller size $n_f$, but the network always operates on input vectors of full size $n_c$, reshaped into $(N, N)$ tensors. Therefore, to evaluate $\tilde{\mathbf{d}} = \mathcal{P}^{\text{net}}(\mathcal{I}, \tilde{\mathbf{r}})$, $\tilde{\mathbf{r}}$ is first padded by inserting zeros into locations corresponding to cells in $\Omega_a$ and $\Omega_s$, and then those locations of the output are removed to obtain $\tilde{\mathbf{d}} \in \mathbb{R}^{n_f}$.

### 4.2.1 ARCHITECTURE

Our neural network architecture (Figure 1) is inspired by geometric multigrid, aiming to propagate information across the computational grid faster than the one-cell-per-iteration of unpreconditioned CG. The architecture is constructed recursively, consisting of levels $1 \leq \ell \leq \mathcal{L}$. A given level $\ell$ operates on an input image $\mathcal{I}^{(\ell)}$ and input vector $\mathbf{r}^{(\ell)}$. It performs a special image-dependent convolution operation on $\mathbf{r}^{(\ell)}$ and then downsamples the resulting vector $\mathbf{y}^{(\ell)}$, as well as $\mathcal{I}^{(\ell)}$, to the next-coarser level $\ell + 1$ using average pooling (analogous to restriction in multigrid). The output of the level $\ell + 1$ subnetwork is then upsampled (analogous to prolongation), run through another

convolution stage, and finally linearly combined with $\mathbf{y}^{(\ell)}$ to obtain the output. At the finest level, $\mathcal{I}^{(1)} = \mathcal{I}$ and $\mathbf{r}^{(1)} = \mathbf{r}$, while at the coarsest level only a single convolution operation is performed.

One crucial difference between our network and existing neural solvers like FluidNet (Tompson et al., 2017) is how geometric information from $\mathcal{I}$ is incorporated. Past architectures treat this geometric data on the same footing as input tensor $\mathbf{r}$, *e.g.* feeding both into standard multi-channel convolution blocks. However, we note that $\mathcal{I}$ determines the entries of $A^{\mathcal{I}}$, and so if the convolutions are to act analogously to the smoothing operations of multigrid, really this geometry information should inform the weights of convolutions applied to $\mathbf{r}$. This motivates our use of custom convolutional blocks whose *spatially varying kernels* depend on local information from $\mathcal{I}$.

Each custom convolutional block (at the right corner in Figure 1) at level $\ell$ learns an affine map from a $3 \times 3$ sliding window in $\mathcal{I}^{(\ell)}$ to a $3 \times 3$ kernel $\mathcal{K}^{(i,j)}$. This affine map is parametrized by a weights tensor $\mathcal{W}$ of shape $(3^2, 3, 3, 3)$ and a bias vector $\mathcal{B} \in \mathbb{R}^{3^2}$. Entry $y_{i,j}$ of the block's output is computed as:

$$y_{i,j} = \sum_{a,b=-1}^{1} \mathcal{K}_{a,b}^{(i,j)} x_{i+a,j+b}, \qquad \mathcal{K}_{a,b}^{(i,j)} := \sum_{c=0}^{2} \sum_{l,m=-1}^{1} \mathcal{W}_{3a+b,c,l,m} \mathcal{I}_{c,i+l,j+m}^{(\ell)} + \mathcal{B}_{3a+b}.$$

Out-of-bounds accesses in these formulas are avoided by padding $\mathcal{I}^{(\ell)}$ with solid pixels (*i.e.*, the values assigned to cells in $\Omega_s$) and $\mathbf{x}$ with zeros.

In multigrid, the solutions obtained on the coarser grids of the hierarchy are *corrections* that are added to the finer grids' solutions; likewise, our network includes connections labeled "linear combination" in Figure 1 that mix in upsampled data from the lower level. Our network determines each of the two coefficients in this combination by learning affine functions of the input image defined by (i) convolving $\mathcal{I}^{(\ell)}$ with a (spatially constant) kernel $\overline{\mathcal{K}}$ of shape $(3, 3, 3)$; (ii) averaging to produce a scalar; and (iii) adding a scalar bias $\overline{\mathcal{B}}$. For efficiency, these evaluation steps are fused into a custom linear block (indicated by blue arrows in Figure 1) that implements the formula:

$$z = \overline{\mathcal{B}} + \frac{1}{3^2 n_c} \sum_{i,j=0}^{N-1} \sum_{c=0}^{2} \sum_{l,m=-1}^{1} \overline{\mathcal{K}}_{c,l,m} \mathcal{I}_{c,i+l,j+m}^{(\ell)}.$$

Our custom network architecture has numerous advantages. Its output is a linear function of the input vector (unlike the nonlinear map learned by Kaneda et al. (2023)), making it easier to interpret as a preconditioner. The architecture is also very lightweight: a model with $\mathcal{L} = 4$ coarsening levels has only $\sim 25\text{k}$ parameters. Its simplicity accelerates network evaluations at solve time, critical to make NPSDO competitive with the state-of-the-art solvers used in practice.

We note that our solver is fully matrix free, with $\mathcal{P}^{\text{net}}$ relying only on the image $\mathcal{I}$ of the simulation scene to infer information about $A^{\mathcal{I}}$. Furthermore, since all network operations are formulated in terms of local windows into $\mathcal{I}$ and $\mathbf{r}$, it can train and run on *problems of any size divisible by* $2^{\mathcal{L}}$.

The 3D version of our architecture is a straightforward extension of the 2D formulas above, simply using larger tensors with additional indices to account for the extra dimension, as well as extending the sums to run over these indices.

### 4.2.2 TRAINING

We train our network $\mathcal{P}^{\text{net}}$ to approximate $A^{\mathcal{I}} \backslash \mathbf{b}$ when presented with image $\mathcal{I}$ and input vector $\mathbf{b}$. We calculate the loss for an example $(\mathcal{I}, A^{\mathcal{I}}, \mathbf{r})$ from our training dataset as the residual norm:

$$Loss = \left\| \mathbf{b} - A^{\mathcal{I}} \mathcal{P}^{\text{net}}(\mathcal{I}, \mathbf{b}) \right\|_2.$$

We found the more involved loss function used in Kaneda et al. (2023) not to benefit our network.

Our training data set consists of 183 matrices collected from 10 different simulation scenes, some of domain shape $(128, 128, 128)$ and others $(256, 128, 128)$. For each matrix, we generate 800 right-hand side vectors using a similar approach to Kaneda et al. (2023), but with far fewer Rayley-Ritz vectors. We first compute 1600 Ritz vectors using Lanczos iterations (Lanczos, 1950) and then

generate from them 800 random linear combinations. These linear combinations are finally normalized and added to the training set. To accelerate data generation, we create the right-hand sides for different matrices in parallel; it takes between 0.5 and 3 hours to generate the data for each scene. Since Ritz vector calculation is expensive, we experimented with other approaches, like picking random vectors or constructing analytical eigenmodes for the Laplacian on $\mathcal{D}$ and masking out entries outside $\Omega$. Unfortunately these cheaper generation techniques led to degraded performance.

In each epoch of training, we loop over the matrices of our dataset in shuffled order. For each matrix, we process all of its 800 right-hand sides in batches of 128, repeating five times. The full training process takes 5-7 days on an AMD EPYC 9554P 64-Core Processor with an NVIDIA RTX 6000 GPU. The training and validation losses are computed every five epochs, and we found it beneficial to terminate after 50 epochs.

### 4.2.3 IMPLEMENTATION

We built our network using PyTorch (Paszke et al. (2019)), but implemented our custom convolutional and linear blocks as custom CUDA extensions. The neural network was trained using single precision floating point.

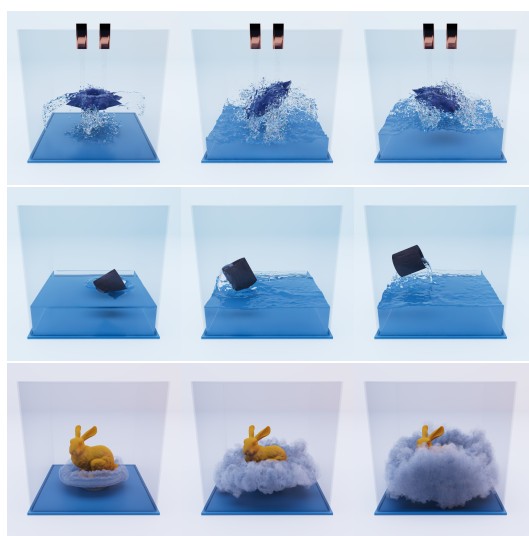

Figure 2: Renderings of some benchmark scenes.

## 5 RESULTS AND ANALYSIS

We evaluate the effectiveness and efficiency of our neural preconditioned solver by comparing it to high-performance state-of-the-art implementations of several baseline methods: unpreconditioned CG provided by the CuPy library (Okuta et al., 2017), as well as CG preconditioned by the algebraic multigrid (AMG) and incomplete Cholesky (IC) implementations from the AMGCL library (Demidov, 2020). All of these baseline methods are accelerated by CUDA backends running on the GPU, with the underlying IC implementation coming from NVIDIA's cuSparse library. Where appropriate, we also compared against past neural preconditioners FluidNet (Tompson et al., 2017) and DCDM (Kaneda et al., 2023). Finally, we included characteristic performance statistics of a popular sparse Cholesky solver CHOLMOD (Chen et al., 2008). In all cases, our method outperforms these baselines, often dramatically.

We executed all benchmarks on a workstation featuring an AMD Ryzen 9 5950X 16-Core Processor and an NVIDIA GeForce RTX 3080 GPU. We used as our convergence criterion for all methods a reduction of the residual norm by a factor of $10^6$, which is sufficiently accurate to eliminate visible simulation artifacts. We evaluate our neural preconditioner in single precision floating point but implement the rest of the NPSDO algorithm in double precision for numerical stability.

We benchmarked on twelve simulation scenes with various shapes—$(128, 128, 128)$, $(256, 128, 128)$, and $(256, 256, 256)$—each providing 200 linear systems to solve. For each solve, we recorded the number of iterations and runtime taken by each solver. These performance statistics are summarized visually in Figures 3-6 and in tabular form in Appendix A.3.

Figure 3a summarizes timings from all solves in our benchmark suite: for each system, we divide the unpreconditioned CG solve time by the other methods' solve times to calculate their speedups and plot a histogram. We note that our method significantly outperforms the others on a majority of solves: ours is fastest on 95.68% of the systems, which account for 98.52% of our total solve time.

Our improvements are more substantial on larger problems, (Figures 3b and c) for two reasons. First, condition numbers increase with size, impeding solvers without effective preconditioners; this is seen clearly by comparing results from two different resolutions (Figures 3d and e). Second, the small matrices $\tilde{A}^{\mathcal{I}}$ correspond to simulation grids with mostly non-fluid cells. While CG, AMGCL

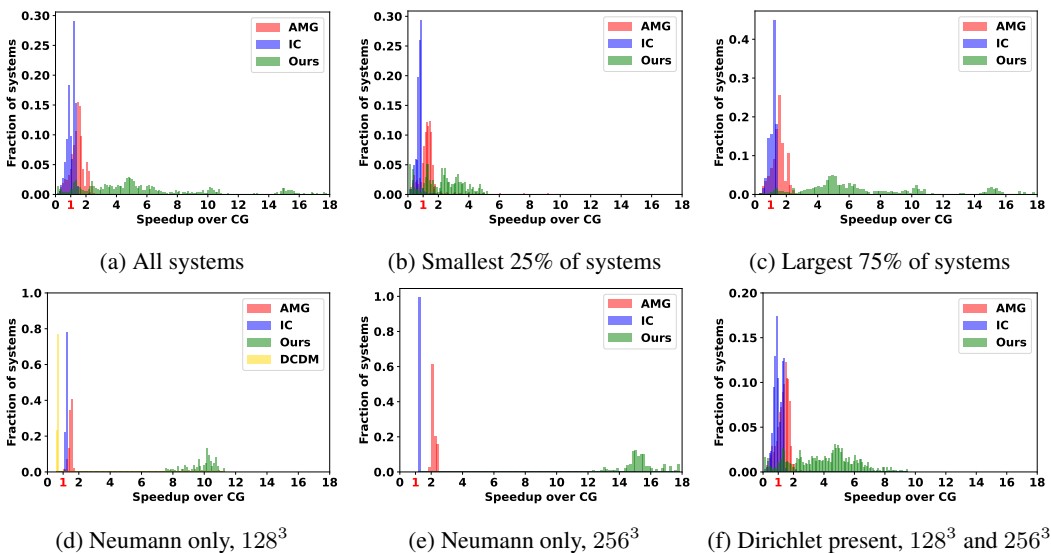

(a) All systems      (b) Smallest 25% of systems      (c) Largest 75% of systems

(d) Neumann only, $128^3$      (e) Neumann only, $256^3$      (f) Dirichlet present, $128^3$ and $256^3$

Figure 3: Histograms of solution speedup vs. a baseline of unpreconditioned CG (a) for all solves; and (b-f) for certain subsets of the systems to help tease apart the different modes of the distribution.

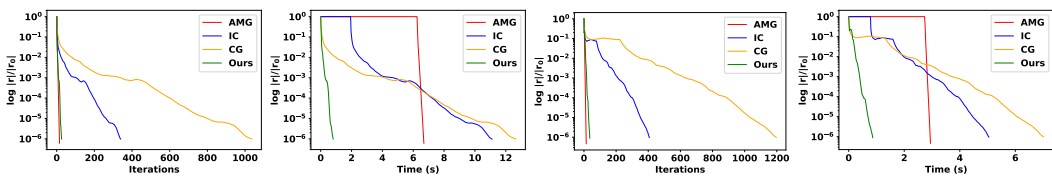

Figure 4: Comparisons among AMG, IC, CG and NSPDO (Ours) on a single frame at $256^3$ with Neumann only BC (left two) and mixed BC (right two).

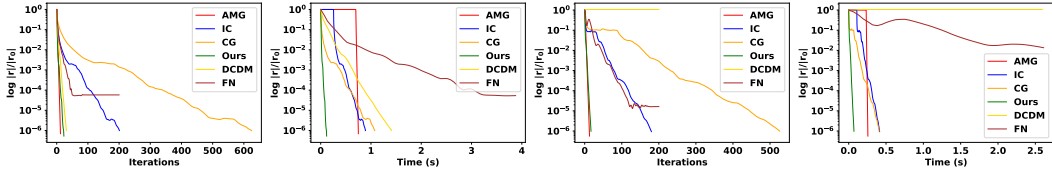

Figure 5: Comparisons among AMG, IC, CG, DCDM, FluidNet (FN) and NSPDO (Ours) on a single frame at $128^3$ with Neumann only BC (left two) and mixed BC (right two).

and IC timings shrink significantly as fluid cells are removed, our network's evaluation cost does not: it always processes all of $\mathcal{D}$ regardless of occupancy. This scaling behavior is visible in Figure 6.

Our speedups are also greater for examples with $\Gamma_d = \emptyset$. DCDM is applicable for these, and so we included in it Figure 3d (but not in Figure 3e due to the network overspilling GPU RAM). DCDM's failure to outperform CG and IC in these results, contrary to (Kaneda et al., 2023), can be attributed to the higher-performance CUDA-accelerated implementations of those baselines used in this work. With Dirichlet conditions (Figure 3f), our preconditioner is less effective, and yet we still outperform the rest on 93.46% of the frames, which account for 97.06% of our total solve time. Statistics are not reported in this setting for DCDM and FluidNet, which struggle to reduce the residual (Figure 5).

Further insights can be obtained by consulting Figures 4 and 5, which show the convergence behavior of each iterative solver on characteristic example problems. AMG is clearly the most effective preconditioner, but this comes at the high cost of rebuilding the multigrid hierarchy before each solve: its iterations cannot even start until long after our solver already converged. Our preconditioner is the second most effective and, due to its lightweight architecture, achieves the fastest solves. DCDM is also quite effective at preconditioning for Neumann-only problems, but its iterations are slowed by costly network evaluations. IC's setup time is shorter than AMG but still substantial, and it is much less effective as a preconditioner.

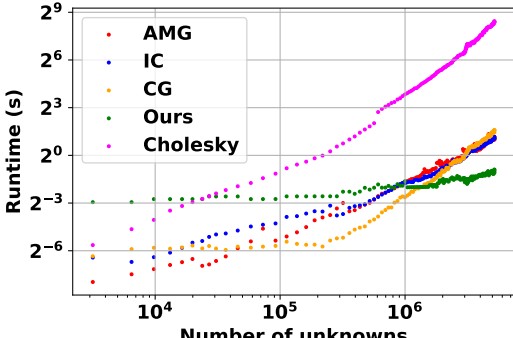

Figure 6: Solver scaling for mixed BC system matrices originating from a fixed-resolution domain ($n_c = 256^3$); matrix row/col size $n_f$ is determined by the proportion of cells occupied by fluid. The vast majority of total solve time is contributed by the high-occupancy systems clustered to the right, where our method outperforms the rest.

We note that the smoke example (Figure 5) also includes a comparison to FluidNet *applied as a preconditioner* for PSDO. In the original paper, FluidNet was presented as a standalone solver, to be run just once per simulation frame. However, in this form it cannot produce highly accurate solutions. Incorporating it as a preconditioner as we do here in theory allows the system to be solved to controlled accuracy, but this solver ended up stalling before reaching a $10^6$ reduction in our experiments; for this reason it was omitted from Figure 3.

On average, our solver spends 79.4% of its time evaluating $\mathcal{P}^{\text{net}}$, 4.4% of its time in orthogonalization, and the remaining 16.2% in other CG operations. In contrast, AMG takes a full 90% of its time in its setup stage. IC's quicker construction and slower convergence mean it takes only 23% in setup. Our architecture also confers GPU memory usage benefits: for $128^3$ grids, our solver uses 1.5GiB of RAM, while FluidNet and DCDM consume 5GiB and 8.3GiB, respectively (Appendix A.3).

## 6 CONCLUSIONS

The neural-preconditioned solver we propose not only addresses more general boundary conditions than past machine learning approaches for the Poisson equation (Tompson et al., 2017; Kaneda et al., 2023) but also dramatically outperforms these solvers. It even surpasses state-of-the art high-performance implementations of standard methods like algebraic multigrid and incomplete Cholesky. It achieves this through a combination of its strong efficacy as a preconditioner and its fast evaluations enabled by our novel lightweight architecture.

Nevertheless, we see several opportunities to improve and extend our solver in future work. First, although we implemented our spatially-varying convolution block in CUDA, it remains the computational bottleneck of the network evaluation and is not yet fully optimized. We are also excited to try porting our architecture to special-purpose acceleration hardware like Apple's Neural Engine; not only could this offer further speedups, but also it would free up GPU cycles for rendering the results in real-time applications like visual effects and games. Second, we would like to explore ways to explicitly enforce symmetry and even positive definiteness of our preconditioning operator so that the less expensive PCG algorithm could be used rather than PSDO. Third, for applications where fluid occupies only a small portion of the computational domain, we would like to develop techniques to exploit sparsity for better scaling (Figure 6). Finally, we look forward to extending our ideas to achieve competitive performance for problems posed on unstructured grids as well as equations with non-constant coefficients, vector-valued unknowns (*e.g.*, elasticity), and nonlinearities.

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

# A  APPENDIX

## A.1  ALGORITHM

For completeness, we provide the following pseudocode for the NPSDO algorithm, which is based on the same iterative solver as DCDM Kaneda et al. (2023) but with our new geometry-aware preconditioner $\mathcal{P}^{\text{net}}$.

---

**Algorithm 1** Neural-preconditioned Steepest Descent with $A$-Orthogonalization (NPSDO)

---

Given matrix $A$, right-hand side $\mathbf{b}$, image $\mathcal{I}$, and trained network $\mathcal{P}^{\text{net}}$
$\mathbf{r}_0 \leftarrow \mathbf{b} - A\mathbf{x}_0$
$k \leftarrow 0$
**while** $\|\mathbf{r}_k\| \geq \epsilon$ **do**
    $k \leftarrow k + 1$
    $\mathbf{d}_k \leftarrow \mathcal{P}^{\text{net}} \left( \mathcal{I}, \frac{\mathbf{r}_{k-1}}{\|\mathbf{r}_{k-1}\|} \right)$
    **for** $k - n_{\text{ortho}} \leq i < k$ **do**
        $\mathbf{d}_k \leftarrow \mathbf{d}_k - \frac{\mathbf{d}_k^\top A \mathbf{d}_i}{\mathbf{d}_i^\top A \mathbf{d}_i} \mathbf{d}_i$
    **end for**
    $\alpha_k \leftarrow \frac{\mathbf{r}_{k-1}^\top \mathbf{d}_k}{\mathbf{d}_k^\top A \mathbf{d}_k}$
    $\mathbf{x}_k \leftarrow \mathbf{x}_{k-1} + \alpha_k \mathbf{d}_k$
    $\mathbf{r}_k \leftarrow \mathbf{b} - A\mathbf{x}_k$
**end while**
**return** $\mathbf{x}_k$

---

## A.2  ALGORITHM COMPARISONS

The following table compares the performances of various iterative solvers preconditioned by $\mathcal{P}^{\text{net}}$. Statistics for unpreconditioned CG are also included for reference. While PCG and Flexible PCG both perform reasonably, PSDO achieves a modest speedup over them.

Table 1: Performance statistics for several solver variants averaged over 200 frames from one scene. PSD is implemented by Algorithm 1 with $n_{\text{ortho}} = 0$. Each solver except unpreconditioned CG (left) was limited to 100 iterations per frame.

|  | CG | PCG | Flexible PCG | PSD | PSDO |
|---|---|---|---|---|---|
| Iterations | 749.21 | 26.075 | 21.565 | 64.475 | 20.67 |
| Time | 2.593 | 0.5507 | 0.4613 | 1.3493 | 0.4502 |

## A.3 TIMING BENCHMARKS

This section lists the average iteration count and runtime for each test case.

Table 2: Average iteration count and runtime across all frames for each test suite

| Examples | AMG Iteration | Time | IC Iteration | Time | CG Iteration | Time | Ours Iteration | Time |
|---|---|---|---|---|---|---|---|---|
| Smoke solid $128^3$ | 11.2 | 0.696 | 196.1 | 0.852 | 612.5 | 1.050 | 21.3 | 0.110 |
| Smoke solid $256^3$ | 14.3 | 6.433 | 343.7 | 11.1 | 1076.8 | 13.245 | 27.9 | 0.870 |
| Smoke bunny $128^3$ | 11.3 | 0.713 | 200.5 | 0.862 | 615.8 | 1.048 | 20.9 | 0.107 |
| Smoke bunny $256^3$ | 14.3 | 6.046 | 345.7 | 10.9 | 1069.7 | 13.042 | 27.5 | 0.857 |
| Scooping $128^3$ | 12.2 | 0.170 | 128.1 | 0.270 | 392.8 | 0.234 | 13.1 | 0.054 |
| Scooping $256^3$ | 12.2 | 1.643 | 246.0 | 1.968 | 749.2 | 2.593 | 20.7 | 0.450 |
| Waterflow torus $128^3$ | 9.4 | 0.086 | 66.9 | 0.129 | 208.6 | 0.095 | 11.7 | 0.044 |
| Waterflow torus $256^3$ | 10.4 | 1.024 | 130.0 | 0.946 | 401.2 | 1.121 | 16.5 | 0.341 |
| Waterflow ball $128^3$ | 11.2 | 0.173 | 136.5 | 0.275 | 405.4 | 0.257 | 17.5 | 0.070 |
| Waterflow ball $256^3$ | 12.1 | 2.442 | 328.4 | 3.559 | 969.7 | 4.875 | 34.3 | 0.795 |
| Dambreak pillars $256 \cdot 128^2$ | 11.2 | 0.476 | 167.7 | 0.625 | 523.2 | 0.704 | 16.5 | 0.113 |
| Dambreak bunny $256 \cdot 128^2$ | 11.1 | 0.395 | 148.2 | 0.514 | 452.1 | 0.522 | 21.6 | 0.144 |
| Average | 11.7 | 1.691 | 203.2 | 2.667 | 623.1 | 3.232 | 20.8 | 0.330 |

## A.4 MEMORY USAGE

The following peak memory usage statistics were recorded with the command `nvidia-smi`.

Table 3: Peak memory usage on a smoke simulation.

| Resolution | AMG | IC | CG | NPSDO | DCDM | FN |
|---|---|---|---|---|---|---|
| $128^3$ | 1248 MiB | 1668 MiB | 1418 MiB | 1548 MiB | 8532 MiB | 5170 MiB |
| $256^3$ | 5032 MiB | 8214 MiB | 3716 MiB | 4776 MiB | NA | NA |

### A.5 ADDITIONAL HISTOGRAMS

The following histograms offer additional views into the data presented in Figure 3, focusing on the linear systems arising from simulations with mixed boundary conditions (i.e., featuring both Dirichlet and Neumann conditions).

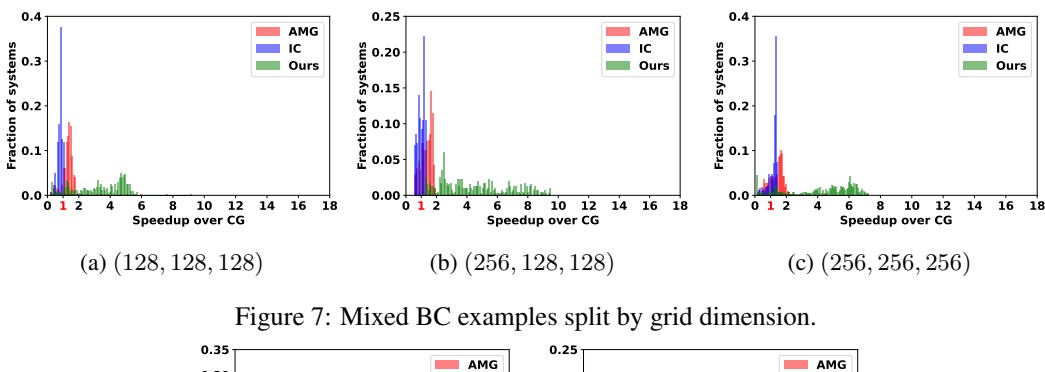

(a) $(128, 128, 128)$      (b) $(256, 128, 128)$      (c) $(256, 256, 256)$

Figure 7: Mixed BC examples split by grid dimension.

(a) Smallest $25\%$      (b) Largest $75\%$

Figure 8: Mixed BC examples split by linear system size into smallest $25\%$ and largest $75\%$.

### A.6 NEW GRID SIZES

The following plots demonstrate the ability of our solver to generalize to domains of different resolutions without retraining the preconditioner: it maintains consistently strong convergence behavior when applied to simulations with grid dimensions not seen during training. We note that $256^3$ resolution simulations reported on in the main paper also were not present in the training set.

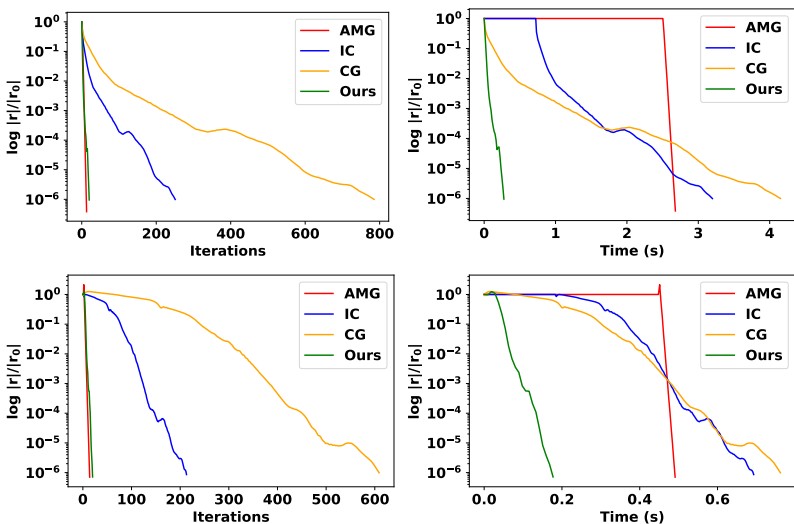

Figure 9: Convergence on a representative frame of a $192^3$-resolution Neumann-only simulation (top pair) and a $(384, 128, 128)$-resolution mixed BC simulation (bottom pair).

### A.7 ANALYSIS OF SYMMETRY

To analyze the deviation of our preconditioner from symmetry we applied our network to pairs of test vectors $\mathbf{x}$ and $\mathbf{y}$ and evaluated the following relative asymmetry measure:

$$\mathcal{S}(\mathbf{x}, \mathbf{y}; \mathcal{I}) := \frac{\left|\mathbf{x} \cdot \boldsymbol{\mathcal{P}}^{\text{net}}(\mathcal{I}, \mathbf{y}) - \mathbf{y} \cdot \boldsymbol{\mathcal{P}}^{\text{net}}(\mathcal{I}, \mathbf{x})\right|}{\sqrt{\left|\mathbf{x} \cdot \boldsymbol{\mathcal{P}}^{\text{net}}(\mathcal{I}, \mathbf{x})\right| \left|\mathbf{y} \cdot \boldsymbol{\mathcal{P}}^{\text{net}}(\mathcal{I}, \mathbf{y})\right|}}.$$

We did this using 100 test vector pairs for each system (image) in our training set, reporting the aggregate statistics in Figure 10a and Figure 10b for two different types of test vectors: (a) random pairs of vectors from the training set, and (b) fully random vectors from a normal distribution. Not only is the trained network's asymmetry small, it is several orders of magnitude smaller than that of the initial network, suggesting that training the network to approximate the inverse of the discrete Laplace operator (a symmetric matrix) naturally promotes symmetry.

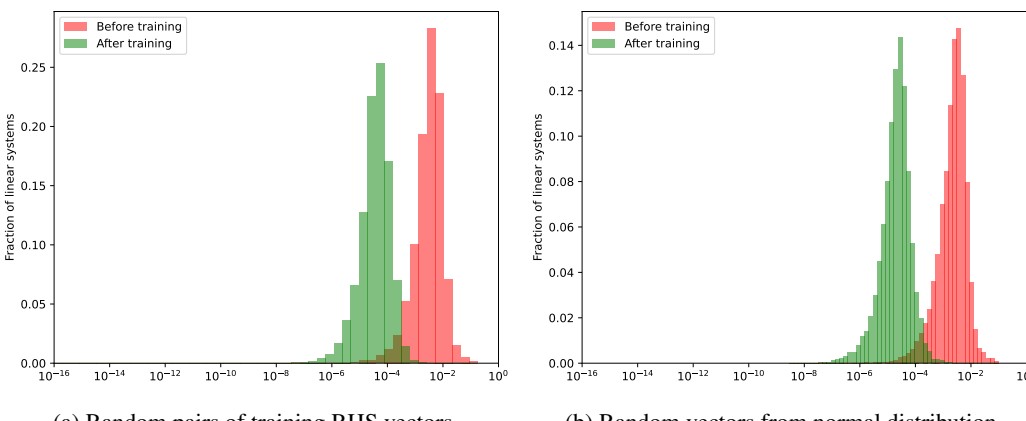

(a) Random pairs of training RHS vectors.  (b) Random vectors from normal distribution.

Figure 10: Histograms of relative symmetry metric $\mathcal{S}(\mathbf{x}, \mathbf{y}; \mathcal{I})$ across 100 test vector pairs $(\mathbf{x}, \mathbf{y})$ for each image $\mathcal{I}$ in the training set.

### A.8 CAPACITY FOR RESIDUAL REDUCTION

To confirm that our solver can achieve high accuracy despite the single-precision arithmetic used in evaluating $\boldsymbol{\mathcal{P}}^{\text{net}}$, we disabled the convergence test and ran for a fixed number of iterations (100), recording the final relative residual achieved for each system in our test set in Figure 11. The median relative residual is $4.01 \times 10^{-15}$.

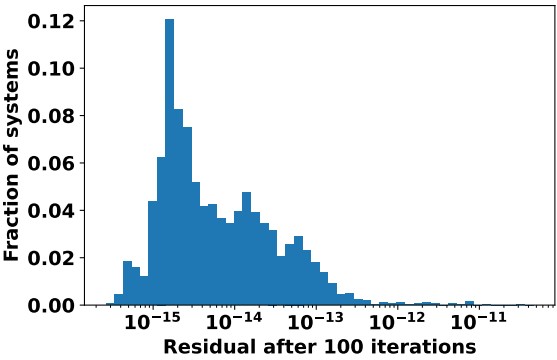

Figure 11: Histogram of the final relative residual achieved after running 100 iterations without a convergence test.

