# OpenReview forum: "A Neural-preconditioned Poisson Solver for Mixed Dirichlet and Neumann Boundary Conditions"
_ICLR.cc/2024/Conference — Submitted to ICLR 2024_

### Official Review · Reviewer_ND3n · 2023-10-22

**Soundness:** 3 good
**Presentation:** 3 good
**Contribution:** 2 fair
**Rating:** 5
**Confidence:** 3

**Summary:**

The manuscript uses a neural network to construct a preconditioner for solving Poisson equations on various geometries with varying boundary conditions. The neural network takes 5-7 days to train before it is a reasonable preconditioner across various geometries. It then provides a modest speed-up over some of the existing preconditioning techniques. The biggest gap in the manuscript is that the preconditioner is not symmetric positive definite, and a highly nonstandard iterative method is advocated that will be challenging to develop a general convergence theory. The multigrid preconditioner remains a good choice for fixed geometries, but here, the neural network preconditioner seems beneficial when applied to a range of geometries.

**Strengths:**

The Deep Conjugate Direction Method (DCDM) is a novel approach that leverages deep learning to approximate the solution of large, sparse, symmetric, positive-definite linear systems of equations. This manuscript is improving on this solver for the setting of Poisson on various geometries. The practical simulations (supplementary) are relatively impressive, demonstrating the effectiveness of their preconditioner.

After training, the constructed preconditioner for Poisson's equation is surprisingly effective at reducing the number of iterations of the DCDM-variant for various geometries.

**Weaknesses:**

Without a positive definite preconditioner, it is likely that the simulations in Figure 2 become unstable (or unphysical) if run for long enough. I think the authors should carefully consider both the computational cost per time step and the numerical stability of the time-stepping scheme when using various preconditioners. This direction is mentioned as a future work, but I think the paper's contribution is compromised without ensuring symmetric positive definiteness.

The lack of theoretical convergence analysis of the preconditioner and iterative solver means that the solver is of limited general use. Since the preconditioner is not positive definite, I suspect that the theoretical convergence of the iterative scheme is extremely difficult to understand.

Training the neural network with a large memory requirement takes a very long time before it can be successfully used in time simulations.

**Questions:**

Can the method be used to construct a neural network-based positive definite precondition? If so, then it at least takes into consideration that the original problem is symmetric, positive-definite linear systems of equations.

---

> ### Author Response · Authors · 2023-11-15
>
> We appreciate the acknowledgement of the surprising effectiveness of our preconditioner. We reiterate our point from the main post that symmetry and positive definiteness, while likely to bring modest benefits, are not critical for our solver's efficacy--or the simulation's stability--in practice.
>
> Regarding the lack of theoretical convergence analysis and the high resource requirements for training, we note that these same limitations apply to the seminal work [Tompson et al. 2017] that has nonetheless made a large impact due to its practical benefits. Especially for the applications we target in real-time simulation and visual effects, the single upfront cost of training the network can pay great dividends over the lifetime of the solver's use.

---

> > ### Comment · Reviewer_ND3n · 2023-11-22
> > **Reply to the official comment by authors**
> >
> > Thank you very much for the authors' reply. However, I still largely disagree with the statement that "symmetry and positive definiteness, while likely to bring modest benefits, are not critical for our solver's efficacy--or the simulation's stability--in practice."
> >
> > For the lack of theoretical convergence analysis, while another work suffers from the same limitation, does not justify the case here. In my opinion, when it comes to preconditioning, the lack of theory (even for some simple problems) here is a significant limitation.
> >
> > Therefore, I decided to keep my score.

---

### Official Review · Reviewer_ddbQ · 2023-10-30

**Soundness:** 3 good
**Presentation:** 3 good
**Contribution:** 3 good
**Rating:** 5
**Confidence:** 4

**Summary:**

This paper proposes a neural network architecture inspired by multigrid to learn an approximate inverse of the discrete Laplace operator, which serves as a preconditioner for an iterative PDE solver. The key contributions are: 1) Handling mixed Dirichlet and Neumann BCs, which prior works cannot; 2) A novel network structure that incorporates boundary information through spatially-varying convolutions; 3) State-of-the-art performance vs AMG, IC and other baselines on fluid simulation examples.

**Strengths:**

1. The network design that encodes boundary information through spatially-varying convolutions is novel and well-motivated. This likely contributes significantly to the method's success in addressing mixed BCs.

2. Comprehensive empirical evaluation on simulation benchmarks demonstrates clear superiority over optimized baseline methods on problems. Statistical analysis provides convincing evidence of the method's benefits.

**Weaknesses:**

1. While the spatially-varying convolutions encode boundary data effectively, their implementation as CUDA kernels is noted to be a computational bottleneck. Further optimization could yield additional speedups.

2. Enforcing symmetry and positive-definiteness of the preconditioning operator was not achieved, limiting the method to a generalization of CG instead of CG itself.

3. The network does not yet leverage sparsity, so may not scale gracefully to extreme sparse problems with many empty grid cells.

4. Only isotropic problems were considered - extending the approach to anisotropic or nonlinear problems is an open question.

5. Training data generation from Lanczos vectors is costly. Finding cheaper alternatives while maintaining accuracy could be valuable for certain applications.

6. Proving theoretical convergence properties like those of standard multigrid would strengthen understanding, though difficult given the learned nature.

7. Quantitative ablation of design choices like coarsening levels, memory usage, etc. could provide further insights.

8. While extensive, the benchmark set considers a subset of potential problem domains - generalizing to new classes would reinforce claims.

**Questions:**

1. Can you exploit sparsity directly in the network/algorithm to better handle extremely sparse problems that arise in practice

2. Can you investigate enforcing symmetry and positive-definiteness of the preconditioner to allow use of standard CG?

3. Can you explore non-Euclidean/anisotropic problem settings to broaden the method's scope?

4. Can you develop cheaper training data generation techniques or consider self-supervised learning alternatives?

5. Can you perform sensitivity studies on architectural hyper-parameters like coarsening levels, memory usage, etc?

6. Can you attempt theoretical analysis of convergence properties to strengthen understanding of the method?

7. Can you broaden empirical evaluation to new problem classes to further validate generalization abilities?

I would like to improve my score if the concerns above are well addressed.

---

> ### Author Response · Authors · 2023-11-15
>
> ## Future Work
> We agree that all of the extensions raised here are interesting avenues for future work, and we are especially excited to see how our spatially-varying kernel architecture performs for problems with heterogeneous and anisotropic constitutive laws. However, we do not see any of these investigations as essential to the completeness and impact of our paper.
>
> ## Exploitation of Sparsity
> Our systems feature sparsity of two types: sparsity of the finite difference Laplacian matrix, and sparsity of the fluid domain occupancy. Matrix sparsity is already exploited by the linear algebra operations in our NPSDO solver (and by our use of small convolution kernels in our network), while domain sparsity is what we mentioned as future work. We note that in typical fluid simulations, the domain geometry is constantly evolving and can range from highly sparse to fully dense; we anticipate inventing a single architecture that works well in both occupancy regimes to be very challenging. Furthermore, efficient algorithms and implementations for sparse convolutions--especially with variable sparsity patterns--are still an active area of research with limited support in available libraries. Even support for sparse matrix formats like CSR and CSC is still in beta stage in PyTorch.
>
> ## Further Optimizations
> One easy optimization that we have already implemented is caching the mixing weights computed by the blue arrows in Figure 1: since these depend only on the input image and not the input residual vector, they do not change after the first NPSDO iteration for a given system. Implementing this caching led to an average speedup of 10.2% across all systems in our test set. The statistics in the updated PDF we have uploaded now reflect this improvement.
>
> ## Self-supervised Learning and Cheaper Training Data Generation
> Our training arguably is already self-supervised, since our loss function does not require a ground-truth solution for each linear system. As mentioned in the paper, we did pursue several less costly approaches for generating the training data that unfortunately did not work well enough. Our existing approach still could be accelerated by using a higher-performance implementation for generating the Ritz vectors, but we do not consider the current implementation prohibitively expensive, especially when compared to training time.
>
> ## Hyperparameter Study
> We will include performance statistics for networks with a range of coarsening levels $\mathcal{L}$ trained using the strategy described in our response to the second review.

---

### Official Review · Reviewer_2G4U · 2023-10-31

**Soundness:** 3 good
**Presentation:** 3 good
**Contribution:** 2 fair
**Rating:** 5
**Confidence:** 4

**Summary:**

The paper presents an approach to construct preconditioner for Poisson equations based on neural networks. The architecture is to mimic the geometric multigrid. The approach has the capability to handle arbitrary shape of the fluid and mixed boundary conditions embeded in a rectangle or cube.

**Strengths:**

The neural network can effectively approximate the inverse of the stiffness matrix from the Poisson equation such that the resulting iterative solver is much faster than the existing methods.

**Weaknesses:**

- The primary contributions of the paper, including the understanding of convolution as a smoother, pooling as the restriction, and the connection between multigrid and convolutional neural networks, have been extensively explored in the following reference:

  Juncai He and Jinchao Xu. "Mg-Net: A unified framework of multigrid and convolutional neural network." arXiv:1901.10415, 2019.

- It appears that the comparison in the paper is limited to iterative solvers and does not account for the time required to train the preconditioner. This approach may not provide a fair assessment. It's worth noting that the setup time for AMG can be comparable to the training time. If one considers the extended matrix on the uniform grid, a fixed hierarchy can be used, and the learning task could focus on the prolongation and restriction operators for different $\mathcal I$. The approach outlined in the following paper may offer valuable insights:

  Alexandr Katrutsa, Talgat Daulbaev, Ivan Oseledets. "Deep Multigrid: learning prolongation and restriction matrices." [arXiv:1711.03825](https://arxiv.org/abs/1711.03825).

**Questions:**

- The data generation process described on the bottom of page 6 needs further clarification. It's not entirely clear whether it involves variations in the domain geometry. If it does, it would be helpful to specify the nature of the randomness associated with the geometry. Additionally, the presentation seems to only consider two types of discretization sizes. If that's the case, it's important to explain how this approach can be applied to systems with different discretization sizes. Does the neural network need to be retrained for each new size? Furthermore, more details about the randomness involved in generating the right-hand side of the equations would be appreciated.
- Could the authors provide an explanation of what the neural network truly learns for an iterative solver to maximize its performance? Specifically, does it focus on learning the smoother or the prolongation operator, or both? Clarity on this aspect would be valuable for understanding the role of the neural network in enhancing solver performance.
- Is it necessary to fix the levels of coarsening in the proposed approach?

---

> ### Author Response · Authors · 2023-11-15
>
> ## Primary Contribution
> We consider our primary contributions to be our novel network architecture and our demonstrations that it can achieve state-of-the-art performance on problems with mixed boundary conditions. We agree that past work has understood the connections between components of traditional CNN architectures and their counterparts in a multigrid hierarchy.
>
> ## Comparisons Limited to Iterative Solvers
> Figure 6 compares against the popular sparse direct solver CHOLMOD.
>
> ## What Does the Network Learn?
> The network learns by optimizing the parameters defining (i) our spatially varying convolution kernel (i.e., the smoother) and (ii) the fixed convolution kernels that compute coefficients for mixing in the interpolated corrections. It does not learn restriction or prolongation operators, which are implemented by fixed pooling and upsampling blocks.
>
> ## Number of Coarsening Levels
> An instance of our network architecture is defined by its number of coarsening levels. This is a hyperparameter that is fixed to 4 in all examples. However, the weights of a network with $\mathcal{L}$ levels can be copied to a new network with $\mathcal{L + 1}$ levels to initialize its training, and we found this strategy beneficial for training deeper hierarchies; we will discuss this approach in the revised draft.
>
> ## Data Generation Process
> The systems comprising our training data set were collected from actual fluid simulations, featuring a diversity of fluid domain shapes.

---

> > ### Comment · Reviewer_2G4U · 2023-11-21
> >
> > I appreciate the authors' detailed explanation; however, I remain unconvinced about the significance of the current work, especially considering the potential for implementing an efficient geometric multigrid algorithm.
> >
> > The numerical example indeed underscores the effectiveness of algebraic multigrid methods. Nevertheless, a common criticism of AMG is the setup time, as evident from the flat period in the AMG curve in Fig 4 and 5.
> >
> > Considering the matrix extension to a larger one associated with a uniform grid, a fixed hierarchy of Geometric Multigrid (GMG) can be applied. The only variable is the smoother, contingent on the boundary conditions. The spatially varying kernels align precisely with the smoothers used in multigrid methods. Implementing geometric multigrid on the extended rectangular domain with Gauss-Seidel smoother can potentially yield a similar behavior. This approach eliminates the need to learn additional smoothers, as they can be determined by the modified stencil.
> >
> > The strategy of embedding the irregular shape of the domain into a rectangular domain and leveraging the structured nature of the latter with a uniform grid is not a new concept in scientific computing. A relevant work in this regard is:
> >
> > "A parallel fictitious domain multigrid preconditioner for the solution of Poisson’s equation in complex geometries"
> > (K.M. Singh, J.J.R. Williams, Computer Methods in Applied Mechanics and Engineering, Volume 194, Issues 45–47, 1 November 2005, Pages 4845-4860).

---

### Official Review · Reviewer_TVSw · 2023-10-31

**Soundness:** 3 good
**Presentation:** 3 good
**Contribution:** 3 good
**Rating:** 6
**Confidence:** 5

**Summary:**

This paper proposed a new data-driven preconditioner for Poisson problems with mixed boundary conditions with a MAC discretization. The PDE arises from the moving boundary of a multiphase flow, thus traditional AMG is unfavorable in terms of the MG hierarchy rebuilding. The key method is built upon the work in Kaneda et al ICML 2023. The authors also tackles a difficulty through the optimizer that the DNN preconditioning approach is non-symmetric, which achieves better result than approaches such as `BLOPEX` in hypre. From the perspective of someone who worked in the business of traditional multigrid solvers/preconditioners for PDEs, the PDE discretization is nothing new, the methodology used is pretty predictable, and there are a few unclear things, however, I still think this is overall a solid work.

**Strengths:**

- In the context of traditional approaches of multigrid, either solver or preconditioner, either geometric or algebraic, imposing (nonhomogeneous) Neumann BC can be challenging and is usually an ad-hoc business.
- The study, from preconditioning pov, is quite well-motivated by introducing the ever-changing BCs for the Poisson problem through the multiphase flow, since traditional AMG has to rebuilt the AMG hierarchy each time step.
- The phase variables as channels are neat practices, which is nicely motivated by the ever-changing phase in simulation. But references should be given (e.g., arXiv:1707.03351).
- The spatially varying kernel (or instance-dependent kernel) is an extremely nice practice, and rightfully so motivated by the mathematical nature of the problem. This is highly connected to attention (even though it is nonlinear) applied to PDE problems (where attention is viewed as a kernel integral), please consider giving a few references in this regard.
- I ran the code myself, and was amazed by the excellent reproducibility of it. However, the instruction of how to install `AMGCL` is problematic in `README.md` for someone with existing header files for `AMGCL`.

**Weaknesses:**

- Personally, I am quite uncomfortable to impose Dirichlet boundary conditions on the pressure variable for the fluid problems among many formulations I have played with for NSE or Stokesian flow. While the temporal discretization of NSE presented on page 3 is a standard splitting scheme (aka "projection method" or "pressure-correction scheme"), the reference given is likely not the right one. In Chorin's 1967 JCP paper, he proposed the famous pressure marching scheme to impose divergence free condition at $(n+1)$-th time step (implied by (4) in the paper), but did not solve a PDE for pressure. I checked Temam's book as well as J. Shen (Temam's student)'s review paper on projection methods, and the scheme featured in this paper is probably attributed to Chorin's 1968 Math. Comp. paper (equation (21) therein for pressure). However, therein, only the Neumann BC is imposed for the pressure. So please give the reference, preferably with equation numbers that where does the Dirichlet part comes from.
- The current presentation of the overall iterative procedure is unclear in section 4. For example, in Figure 1, only $\mathbf{r}\mapsto \mathcal{P}^{\text{net}}(\mathcal{I}, \mathbf{r})$ is shown. I suggest move A.1 to the main part.
- The biggest weakness is perhaps that the "CNN" used is linear, then it is nothing but a single parametrized convolution, as stacking convolutions without nonlinearity is still a convolution just with a different kernel sizes.
- Maybe this is up to debate, that the authors said "AMG setting up stage takes too long". I think this is an unfair comparison, on a fixed mesh, AMG hierarchy has to be set up only once, and for GMG it is automatic. Moreover, AMG setting up is automated on any mesh and (mostly) robust for specific PDEs that needs no training stage. Another counterpoint would be that the current approach (based on CNN) is limited to Cartesian mesh-based discretization such as MAC.

### Minor things
- I suggest the authors rewrite the abstract. As writings like "The Poisson equation is ubiquitous in scientific computing: it governs... The most popular Poisson discretizations yield large sparse
linear systems" are more appropriate in the introduction than the abstract, a better way to put things in the contexts is just saying large systems arising from PDE's discretization is hard to solve.
- The acronym DCDM goes undefined across the paper.
- Page 1: "matrix-norm" should be "matrix norm".
- Page 2: "their loss functions is...".
- Page 2: "disretizations" -> discretizations.
- Page 5: if $P = LL^{\top}$ is the preconditioner (assuming the new system is $PA\mathbf{x} = P\mathbf{b}$), then CG steps build subspaces that are $L^{\top}A L$-orthogonal, not $L^{-1} A L^{-\top}$-orthogonal (to converge to $L^{-1}\mathbf{x}^*$ for $\mathbf{x}^* = A^{-1}\mathbf{b}$).
- Personally, I am not in favor of referring $\mathcal{I}$ an "image" in this context because one may confuse this term with the image of an operator. Of course, it is up to the authors' judgement on this.

**Questions:**

- The targeted audience who works in the solver business would usually not know what "PIC/FLIP blend transfer scheme" is, I suggest elaborate it somewhere in the context of this paper, especially considering a similar acronym IC stands for incomplete Cholesky.
- Judging by the name of Harlow 1964 Meth. Comp. Phys. paper, it is the ref for PIC? The correct origin of MAC should be Harlow and Welch's 1965 paper in The Physics of Fluids, some people like to cite this tech report as well (https://www.osti.gov/biblio/4563173).
- Page 4: "The PSDO algorithm can be understood as a modification of standard CG that replaces the residual with the preconditioned residual as the starting point for generating search directions and, consequently, cannot enjoy many of the simplifications baked into the traditional algorithm". This sentence is somewhat too long in reviewer's humble opinion, and up to the authors' judgement, I suggest rewrite it a little. Meanwhile, please consider spending a few words to explain the reason why PSDO cannot achieve CG's automatic $A$-conjugate directions. It says that PSDO does "an exact line search", so it does not a subspace search like the CG does?
- Page 5-6, the default behavior of upsampling using the interpolation in `nn.functional` is `nearest` which gives a piecewise constant function that is unfavorable in this context, please test `bilinear` which is MG prolongation.
- The single-precision for NN and double for the CG is a good practice. However, it would be interesting to see whether this combination can reach relative residual `1e-8` (cf. those in Figure 5).

---

> ### Author Response · Authors · 2023-11-15
>
> ## Dirichlet Boundary Conditions for Free Surfaces
> We note that our application is *inviscid* fluid simulation, where the omission of viscous stress means that the surface pressure simply equals the applied pressure (e.g., see the section entitled "Boundary Conditions at Free Surface" in [Harlow and Welch 1965], or page 15 of the textbook [Bridson 2015: Fluid Simulation for Computer Graphics]); this is naturally posed as a Dirichlet condition.
>
> ## Equivalence to One Large Convolution
> Indeed, because the network output is a linear function of the input vector, it can be interpreted as a single convolution with one large spatially varying kernel (whose size depends on the number of layers) that is parametrized by our network weights. We do note that the network output is a nonlinear polynomial function of the input *image* since the coefficients of successive kernels end up multiplying each other. We see the simplicity and linearity of our network not as a weakness but as a contribution (showing a heavier-weight, nonlinear architecture is unnecessary) and a practical advantage (accelerating inference at runtime).
>
> ## Bi/trilinear Prolongation
> Thank you for pointing this out! Yes, trilinear interpolation is a closer analogue for MG prolongation. Simply substituting it for the default upsampling in our old trained network achieves a measurable speedup (reducing the iteration count across all linear systems in our test set by an average of 5.3%). We have updated the statistics reported in our paper to reflect these improvements. We have also begun training a new model incorporating this trilinear interpolation, which we expect to achieve further improvements.
>
> ## A-conjugacy of Directions
> Despite using an exact line search as in PCG, enforcing conjugacy with just the previous search direction does not automatically ensure conjugacy with all past directions. This property enjoyed by CG hinges on symmetry, and we would be happy to provide detailed proofs in the appendix.
>
> ## Tolerance Achievable
> The accuracy of our solver is not limited by the preconditioner's single-precision arithmetic and can easily surpass a relative residual of $10^{-8}$. To test this, we disabled the convergence test and executed each solve in our entire test set for a fixed number of iterations (100). We measured a median relative residual of $4.01\times10^{-15}$ and report a histogram in the newly added Figure 11 of the appendix.

---

> > ### Comment · Reviewer_TVSw · 2023-11-21
> >
> > 1. I've checked what has been revised and studied the numerical experiments in the repo in details one more time;
> > 2. I've been learning two references in more details, Huang et al SISC 2023 and He-Xu Mg-net paper mentioned by Reviewer 2G4U;
> > 3. After learning that an inaccurate inner "preconditioner" bounded by the single precision can still achieve the machine epsilon in double when paired with a CG-like outer iteration, I changed my view of what are the weights are actually learning in this case;
> >
> > I agree with the authors that the output is nonlinear thanks to the indicator mapping being very special, and very likely building a single large convolution (two-grid) paired with restriction won't work (attributing to 3. above). However, due to 2. and 1., I will keep my initial evaluation unchanged. I think this is a solid work, but the new contribution is limited to a very specific case (but I deemed it important), also compared with AMGCL.
> >
> > By the way, I am not sure if I should address this, since the authors already acknowledged that the lacking of a positive definiteness and symmetry as their approach's limitation. I found it kinda pointless to point this out in the reviews (as two other reviewer did). Nevertheless, I am kinda disappointed that the authors did not address the symmetry part. A hint here is that please consider what makes the two-grid propagation formula $
> > (I-R_2 A)^{m_{\text{post}}} (I - P(P^{\top}AP)^{-1}P^T A ) (I-R_1 A)^{m_{\text{pre}}}$ symmetric.

---

### Author Response · Authors · 2023-11-15

We thank all reviewers for their insightful comments; we will incorporate your suggestions and make the requested corrections in an updated draft. In the meantime, we address below some of the broader concerns and will post individual replies to address more specific points.

## Connections Between CNNs and Multigrid (R1, R2)
We agree that past work has identified and exploited strong connections between CNNs and traditional multigrid solvers. However, to the best of our knowledge, we are the first to implement a spatially varying smoothing operator using a soft attention mechanism, which is key to our preconditioner's effectiveness in the face of changing domains/boundary conditions (and, we conjecture, for future extensions to variable-coefficient and nonlinear PDEs); we see this novel feature of our network architecture as our primary technical contribution. We will expand our related work section to discuss these past works and the connections between our spatially varying kernels and attention.

## Fairness of Comparison to MG Setup Time (R1, R2)
Even for fixed meshes, the preconditioner must be rebuilt when numerical values in the matrix change. In practice, this means updating the preconditioner at every frame of our fluid simulation, even if the solver is formulated on the full, fixed uniform grid. While these updates are indeed automatic and robust with AMG, our experiments show them to be a significant bottleneck---to the point that AMG struggles to outperform a highly tuned GPU implementation of unpreconditioned CG in the accuracy regime relevant to real-time simulation. Nonlinear problems that we hope eventually to tackle with a variant of our network architecture are another example where preconditioner updates would be needed despite a fixed mesh: numerical values in the tangent stiffness matrix's change at every Newton iteration.

We disagree that AMG setup time is analogous to our training: AMG setup is specific to a particular system matrix, and it would be impractical to construct and store preconditioners for every system matrix that can be encountered during real-time fluid simulation. On the contrary, our results show that after completing the offline training phase, our solver is effective at preconditioning thousands of previously unseen linear systems with distinct matrices arising from a variety of scenes and resolutions.

## Symmetry and Positive Definiteness of the Preconditioner (R3, R4)
Although we mentioned enforcement of symmetry and positive definiteness as an interesting subject for future investigation, lacking these properties is far from detrimental to our method's performance (especially with the NPSDO algorithm), and we do not see it compromising our paper's contribution. First, we emphasize that positive definiteness of the preconditioner has no bearing on the stability of the fluid simulation: as long as the Poisson equation still can be solved to sufficient accuracy (which our algorithm accomplishes rapidly and robustly in 100% of test cases), the details of the solver do not influence the simulation behavior. Second, the additional linear algebra operations in NPSDO vs PCG necessitated by asymmetry (e.g., the extra A-orthogonalizations) do not add significant overhead compared to the cost of network evaluation, which accounts for approximately 80% of the solver time. Finally, the reasonable, albeit suboptimal, performance of the plain PCG algorithm reported in Table 1 in the appendix suggest that the deviations of our preconditioner from symmetry and positive definiteness are not severe; this is confirmed by experiments newly added to the appendix (Figure 10) that measure the magnitude of symmetry violation of our trained operator on randomly generated vectors. We note that training shrinks these violations to a small relative magnitude, suggesting that training our network to approximate the inverse of a symmetric matrix already guides it toward symmetry.

## Generalization Across Discretization Sizes (R2)
All results reported in the paper were generated by a *single network* that was trained on simulation data from grids of sizes 128^3 and 256x128x128. The same trained model achieves even greater speedups on the more difficult 256^3 grid (Figure 3) despite never seeing training data of that size. A trained model with $\mathcal{L}$ layers is directly applicable to any problem with sizes divisible by $2^\mathcal{L}$. We have included more benchmark statistics in the updated appendix that show strong performance on grid sizes 192^3 and 384x128x128, which again were not present in the training dataset. In theory, the network also can be applied to grids of arbitrary shape by padding with solid pixels (the padding value already used for out-of-bounds reads). For substantially larger grids, it could be beneficial to increase the number of coarsening layers.

---

### Author Response · Authors · 2023-11-20

We thank all the reviewers again for their insightful feedbacks. In the meanwhile, we have posted our comments to address the concerns. Any follow-up discussions are highly appreciated, and will help us further improve our paper!

---

### Meta-Review · Area_Chair_BLHo · 2023-12-06

**Metareview:**

The submission introduces a novel approach that integrates neural networks with classical Poisson solvers to address mixed boundary conditions in fluid simulations. The authors showcase a novel neural network architecture, designed to approximate the inverse of structured-grid Laplace operators, and demonstrate its practical effectiveness in a range of challenging test cases. The empirical results presented in the paper are impressive, illustrating the solver's potential in outperforming traditional methods in specific fluid simulation benchmarks.

However, despite its strong empirical performance, the paper has a number of weaknesses. First, the paper perhaps understates the limitation of needing to train a neural network. The approach requires extensive training, that may limit the practical applicability and scalability of the method, and make it less feasible for broader use in real-world scenarios. One reviewer points out that the problem under consideration could perhaps be better solved using geometric multigrid, with a significant reduction in setup costs. Second, the lack of positive definiteness and symmetry can lead to unpredictable behavior, including the potential for divergence or inefficient convergence, which in turn compromises the reliability and effectiveness of the entire solver.

Regarding the lack of theoretical guarantees, the authors point out that CG is guaranteed to terminate in $n_c$ iterations under exact arithmetic. Moreover, with exact arithmetic, the solver is guaranteed to reach any error level after a sufficient number of iterations. However, both are moot points. First, CG does not terminate in finite iterations, due to the rapid loss of orthogonality in the implied Lanczos basis; see e.g. Greenbaum & Strakos (SIMAX, 1992). Also, the argument that the neural network can at worse return random directions is not very convincing, as it is very expensive to train, and there are much easier and cheaper ways of getting random directions. One can simply run naive gradient descent and obtain an algorithm that will eventually attain any error level above the noise floor, using only finite precision arithmetic.

The paper's strengths in innovation and practical results are acknowledged, but in the end, they do not fully offset these weaknesses.

**Justification For Why Not Higher Score:**

The paper has a number of weaknesses that remain unaddressed after author rebuttals.

**Justification For Why Not Lower Score:**

N/A

---

### Decision · Program_Chairs · 2024-01-16

Reject